# Functional insights of novel Bathyarchaeia reveal metabolic versatility in their role in peatlands of the Peruvian Amazon

Michael J. Pavia,[1,2,3] Arkadiy I. Garber,[1,3] Sarah Avalle,[1] Franco Macedo-Tafur,[4] Rodil Tello-Espinoza,[4,5] Hinsby Cadillo-Quiroz[1,2,3]

**ABSTRACT** The decomposition of soil organic carbon within tropical peatlands is influenced by the functional composition of the microbial community. In this study, building upon our previous work, we recovered a total of 28 metagenome-assembled genomes (MAGs) classified as Bathyarchaeia from the tropical peatlands of the Pastaza-Marañón Foreland Basin (PMFB) in the Amazon. Using phylogenomic analyses, we identified nine genus-level clades to have representatives from the PMFB, with four forming a putative novel family ("*Candidatus* Paludivitaceae") endemic to peatlands. We focus on the *Ca*. Paludivitaceae MAGs due to the novelty of this group and the limited understanding of their role within tropical peatlands. Functional analysis of these MAGs reveals that this putative family comprises facultative anaerobes, possessing the genetic potential for oxygen, sulfide, or nitrogen oxidation. This metabolic versatility can be coupled to the fermentation of acetoin, propanol, or proline. The other clades outside *Ca*. Paludivitaceae are putatively capable of acetogenesis and *de novo* amino acid biosynthesis and encode a high amount of $Fe^{3+}$ transporters. Crucially, the *Ca*. Paludivitaceae are predicted to be carboxydotrophic, capable of utilizing CO for energy generation or biomass production. Through this metabolism, they could detoxify the environment from CO, a byproduct of methanogenesis, or produce methanogenic substrates like $CO_2$ and $H_2$. Overall, our results show the complex metabolism and various lineages of Bathyarchaeia within tropical peatlands pointing to the need to further evaluate their role in these ecosystems.

**IMPORTANCE** With the expansion of the *Candidatus* Paludivitaceae family by the assembly of 28 new metagenome assembled genomes, this study provides novel insights into their metabolic diversity and ecological significance in peatland ecosystems. From a comprehensive phylogenic and functional analysis, we have elucidated their putative unique facultative anaerobic capabilities and CO detoxification potential. This research highlights their crucial role in carbon cycling and greenhouse gas regulation. These findings are essential for resolving the microbial processes affecting peat soil stability, offering new perspectives on the ecological roles of previously underexplored and underrepresented archaeal populations.

**KEYWORDS** Amazon, Bathyarchaeia, soil microbiology, microbial communities, facultative anaerobes, peatland, archaea

Peatlands, a type of wetland, are characterized by their large carbon storage capacity, in which the rates of primary production exceed decomposition rates (1). The slow process of organic matter (OM) decomposition in peatlands is a consequence of several limiting factors, including continuous water-logged soils leading to anoxic conditions (2), lower temperatures compared with surrounding soils from low conductive properties (3), acidification from biological metabolism (4), lower concentrations of nutrients from low

**Peer Reviewer** Bing Song, Danmarks Tekniske Universitet, Copenhagen, Denmark

Address correspondence to Hinsby Cadillo-Quiroz, hinsby@asu.edu.

The authors declare no conflict of interest.

See the funding table on p. 12.

input or depletion (5), and accumulation of phenolic compounds due to the inhibition of phenol oxidases in the absence of oxygen (6).

The Pastaza-Marañón Foreland Basin (PMFB) is home to a large expanse of tropical peatlands (~67,000 km$^2$) in the Amazon and is estimated to contain 3.1 Pg of stored soil carbon (7, 8). In addition to functioning as a terrestrial carbon reservoir, the PMFB emits 3.16 to 41.1 Tg of $CH_4$ per year (9). Understanding the roles of distinct microbial populations in the transformation of stored organic carbon in these peatlands is essential to addressing the projected shift of this region from a carbon sink to a carbon source as a result of climate change (10, 11). Archaea constitute a significant proportion of the microbial community in the soils of the PMFB; most notably the Bathyarchaeia, which range in relative abundance from 3% to 8% in the shallow soils (≤20 cm below surface) and up to 20% of the community in the deeper soils (12, 13). Previously classified as the "Miscellaneous Crenarchaeotal Group," their presence has been detected in multiple terrestrial and aquatic environments, making them a potentially important contributor to sedimentary carbon cycling (14, 15). Despite their potential importance in the carbon cycle, we currently lack axenic cultured representatives, hindering our ability to understand their physiology and functional role within the environment.

Metagenomic approaches have provided valuable information on the environmental distribution and metabolic diversity of Bathyarchaeia (16–20). Metabolic reconstructions predict a broad range of possible metabolisms within members of the Bathyarchaeia including, but not limited to, anaerobic methane oxidation (21), acetogenesis (22), phototrophy (16, 23), heterotrophy based on methylated compounds (19, 24), aromatic compounds (25), or detrital proteins (17, 26, 27). Furthermore, recovery of metagenome-assembled genomes (MAGs) classified as Bathyarchaeia from the Surat Basin (Eastern Australia) detected the potential for methylotrophic methanogenesis; however, these MAGs contain few methanogenic marker genes and mostly "Mcr-like" genes (24, 28). Cultivation-centered studies enriching for Bathyarchaeia have demonstrated their growth on various complex carbon substrates (29–32), as well as the production of acetate from guaiacol (33) so far only growing in mixed cultures. In addition to functioning in the cycling of carbon, the Bathyarchaeia are predicted to play a role in sulfur and nitrogen cycling, as well as in vitamin B12 production (16).

While understanding the distribution and potential environmental drivers of the Bathyarchaeia is important, it remains unclear what metabolisms or functions they may carry out within tropical peatlands, such as the PMFB. Thus, to investigate the putative phylogenetic variation and metabolic potential of Bathyarchaeia in the PMFB region, we evaluated MAGs from shallow and deep peat soil layers from one nutrient-rich and three oligotrophic peatlands with distinct vegetation dominance, respectively: mixed forest (Buena Vista), pole forest (San Jorge), palm swamp (Quistococha), and open (Maquía) peatland (5, 34). Using Bathyarchaeia MAGs from the PMFB plus nearly all publicly available high-quality MAGs, we conducted phylogenomic, pangenomic, and functional analyses to understand (i) the phylogenetic and metagenomic-assembled gene content distribution of PMFB Bathyarchaeia within this diverse phylum and (ii) predict the putative metabolisms unique to the PMFB Bathyarchaeia.

## MATERIALS AND METHODS

### Study sites, metagenomes, and MAG assembly

We evaluated MAGs from metagenomes of four peatlands, previously described (12, 34) within the PMFB: Buena Vista (BVA), San Jorge (SJO), Quistococha (QUI), and Maquía (MAQ). The general characteristics of each site are detailed in Table S1. Earlier sampling and metagenome sequencing for shallow soils in BVA, QUI, and SJO are described in our earlier report (35). Briefly, soils of 0–10 and 10–20 cm were collected between July and October 2015 and January and February 2016, and extracted DNA was sequenced with Illumina Hi Seq 2500 2 × 151 bp technologies at the Joint Genome Institute (JGI). To expand our studies, deep soil samples were collected at 60–100 cm deep in August

2016 for QUI and SJO and in August 2017 for MAQ, transported and stored frozen until DNA extraction using the previously reported method (35). Library preparation and Illumina sequencing were conducted under the same approach at JGI as above, all part of JGI's Community Sequencing Program (proposal: doi:/10.46936/10.25585/60000849) and detailed in File S1.

For MAG assemblies, metagenomic reads were decontaminated, trimmed, and quality score filtered following the Joint Genome Institute IMG protocol (36). Quality control metagenomes were assembled using MEGAHIT (v1.1.3) (37) using the default settings, and quality was assessed with QUAST (v3) (38). MAQ metagenomes were co-assembled by transect, whereas SJO and QUI were a single assembly due to low DNA and low sequencing yield of replicates. Contigs were quality-controlled as described elsewhere (35) and binned using MetaBAT2 (v2.12.1) (39) with a minimum cutoff length of 2,000 bp. The resulting bins were dereplicated and curated using Anvi'o (v6.1) (40). MAGs >50% completeness and <10% redundant as determined by CheckM (41) were classified using GTDB-tk (42) against the Genome Taxonomy Database release 07-RS207. The relative abundance of MAGs was calculated by mapping metagenomes back to MAGs using bowtie2 (43), and the JGI script provided in MetaBAT2 (39) was used to calculate the total contig coverage; total coverage was then normalized by MAG size.

## Phylogenetic inference and average amino acid identity (AAI) comparisons

To determine the distribution of PMFB Bathyarchaeia within the class, we performed phylogenetic inference using an in-group of 238 Bathyarchaeia MAGs, which included both those from this study and publicly available ones (File S2). Prior to phylogenetic analysis, Bathyarchaeia MAGs were filtered for bias based on genome size (>1 Mb), redundancy (<10%), completeness (>50%), and AAI scores greater than 95% from MAGs sampled from the same environment. In instances where more than one MAG from the same environment had an AAI greater than 95%, we considered them replicates and selected the MAG with higher completeness and lower contamination. The outgroup consisted of 37 publicly available genomes representing both Thermoproteota and Euryarchaeota organisms. Anvi'o (v6.1) (44) was used to identify, concatenate, and align 54, single-copy genes (Table S3) that were found in at least 82% (191) of MAGs (as determined from the mean completeness score for just Bathyarchaeia MAGs). The alignment was used for maximum likelihood (ML) inference supported by bootstrap-resampling 300 times using RAxML (v8.2.12) (45) with the PROTCATBLOSUM62 model of amino acid substitution. AAI scores, calculated using github.com/mooreryan/aai, were clustered based on ≥65% similarity, and clusters were assigned an operational taxonomic rank at the genus level, previously proposed by Konstantinidis et al. (46). In this study, we have denoted each putative genus-level cluster as a Bathyarchaeia clade (BC). A chi-squared test was used to detect if there was a significant association between BCs and ecosystem type. GC% and predicted genome size were assessed for significance to BCs with Kruskal-Wallis and Tukey's multiple comparison and were used on BCs that contained more than four MAGs. All differences were considered significant at a $P$-value < 0.05 and visualized with ggstatsplot (47).

## Meta-pangenome and functional enrichment analysis of bathyarchaeia

The Aniv'o pangenomic workflow (44) was used to identify gene clusters within BCs that contained MAGs from the PMFB (nine BCs in total). Briefly, this workflow calculates similarities across all open reading frames, removes weak hits with a chosen minbit heuristic score of 0.6, and uses the MCL algorithm for gene cluster identification; partial genes were excluded (48). Completion and contamination statistics were used for operational cutoffs to designate gene clusters as relaxed-core, shell, cloud, and singletons (Table S4). Gene clusters were annotated using eggNOG (v5) (49) and the nr database (accessed August 2021) (50) using a consensus sequence built from the alignments of each gene cluster. Briefly, the consensus sequence was built on the

frequency of amino acids present at each position, and in positions where there was a tie, a residue was chosen at random.

## RESULTS AND DISCUSSION

### Microbial community composition from assemblies of new MAGs

New assemblies were completed for deeper soil metagenomes from QUI and SJO and shallow soil metagenome from MAQ, followed by genome binning resulting in a total of 122 high-quality (HQ) and medium-quality (MQ) MAGs (51) (Fig. S1). Acidobacteriae and Nitrososphaeria were highly represented in MAGs recovered across all three sites. Moreover, we recovered 14 novel MAGs belonging to the Bathyarchaeia and three HQ MAGs classified as Lokiarchaeia (Fig. S1). Consistent with previous studies utilizing 16S rRNA gene amplicons (12, 13), Bathyarchaeia were found to be abundant in deep peat, accounting for ~15% of the community. This study extends these findings by recovering HQ and MQ MAGs, providing a more detailed genomic representation of the Bathyarchaeia populations within peatlands from QUI, SJO, and MAQ from the PMFB (12, 13).

### Taxonomic placement of PMFB Bathyarchaeia

Bathyarchaeia MAGs from the PMFB (28 in total, seven from the new assemblies passing inclusion criteria) were evaluated in combination with 210 other MAGs from published studies with a median completeness of 81.9% and redundancy 3.3% (detailed in File S2). Phylogenomic clades within the Bathyarchaeia were inferred from the robustness of tree topology based on a 54 single-copy gene (SCG) ML phylogenomic tree in combination with AAI between MAGs (Fig. 1; Fig. S2). Consensus between the phylogenomic tree and AAI suggests that there are 60 distinct clades (operational taxonomic units at the genus level) within our Bathyarchaeia data set. This is consistent with a recent report on the phylogenetic diversity of Bathyarchaeia (52). We note the clade equivalents with (52) but expand the analysis by advancing predictions for novel Bathyarchaeia from Amazon peatlands. Among the Bathyarchaeia clades (BCs), nine contained MAGs from the PMFB, with three comprised exclusively of PMFB MAGs (BC15, BC39, and BC42). Found within BC1 and BC3 (also noted as Baizomonadales) are Bathyarchaeia MAGs from both the PMFB and other environments such as hot springs, termite guts, and permafrost. BC36 (also noted as Houtuarculales), is primarily comprised of MAGs from grasslands, with only one from the PMFB. Although the PMFB MAGs span the diversity of Bathyarchaeia, several shallow branches (BC38 – BC42), within the noted Houtuarculales, were comprised primarily of our tropical peatland populations. BC15, also noted as Baizomonadales, BC39, and BC42, located within the Houtuarculales group, have not been previously reported, thus representing novel lineages of Bathyarchaeia that inhabit the soils of the PMFB.

The distribution of MAGs recovered by environment and the sequence composition among BCs exhibited non-random distribution by source and clade-specific genomic signatures (Fig. 2; Fig. S3). A chi-squared test showed that among the 16 BCs with more than four representative MAGs, 13 BCs are significantly associated with one ecosystem type (Fig. 2). The BCs that show the strongest association with an ecosystem were BC18 (also noted as Baizomonadales), BC5 (also noted as Baizomonadales), and BC47 (also noted as Houtuarculales) in hot springs; BC19 (also noted as Baizomonadales), in hydrothermal vents; BC29, BC8, and BC12 (all in the group also noted as Baizomonadales) in marine zones; BC36 (also noted as Houtuarculales) in grasslands; and BC38 (also noted as Houtuarculales) in peatlands. Other BCs (particularly BC1, BC11, and BC24, all in the group also noted Baizomonadales) are less significant and show a widespread distribution across multiple environments. Previous studies have suggested that Bathyarchaeia are non-randomly distributed along salinity and oxygen concentrations (53, 54), which may be applicable to BC8 and BC24, which were recovered from estuarine environments. Other studies have also indicated that total organic carbon influences lineage

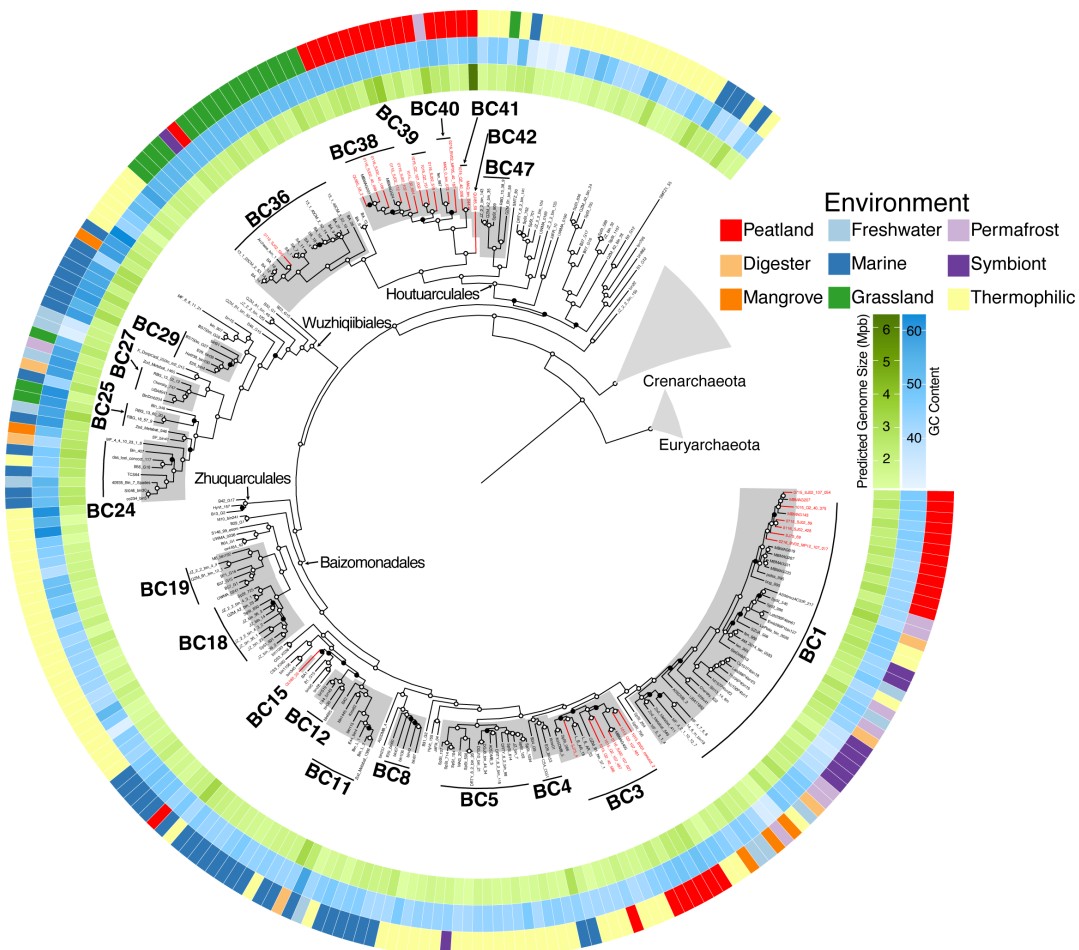

**FIG 1** Maximum likelihood phylogenetic inference of 233 Bathyarchaeia MAGs constructed using 54 concatenated protein sequences. The robustness of MAG placement was assessed using 300 bootstrapping support. Bootstrap support of ≥70% and <70% (black) is represented by hollow and black circles, respectively. Clades with gray backgrounds and that are labeled represent potential genera (based on tree topology and AAI) with more than four MAGs. MAGs recovered from the PMFB are in red. Orders with representatives from the PMFB are labeled at the node. Corresponding heatmaps display the predicted genome size and GC%, for each along with the environmental source from which each MAG was recovered.

distribution (55, 56). In line with these observations, BC11 is primarily comprised of MAGs recovered from estuarine sediment ecosystems including an enrichment culture on lignin.

Specialization in genomic characteristics, such as genome size and GC content, is consistent within archaeal species (57, 58). To assess whether the environment is a predictor of genomic signatures in Bathyarchaeia, we performed a Kruskal–Wallis test on clades containing more than four representative MAGS (Fig. S3). Minimal significant differences were observed in both predicted genome size and GC content between MAGs recovered from different ecosystems, with many being indiscernible from each other. The weak relationship between archaeal genome size and phylogeny has been observed in other groups and may be influenced by the complexity of archaeal genomes (59). Alternatively, we found that GC content can be a strong indicator for BC. Many basal BCs have a GC content within 40%–50%, whereas shallower branching clades (BC25, BC27, BC29, and BC36) have a significantly higher GC%, ranging from 50%–60%. These four clades were also found to be significantly associated with specific ecosystems and may have undergone selective pressures linked to chronic energy stress (60). These clades are primarily recovered from marine sediment, littoral marine zones, and grasslands, which are characteristic of being anoxic with low primary production and are energy-starved ecosystems. Commonly found in hot springs, BC47 has the lowest GC

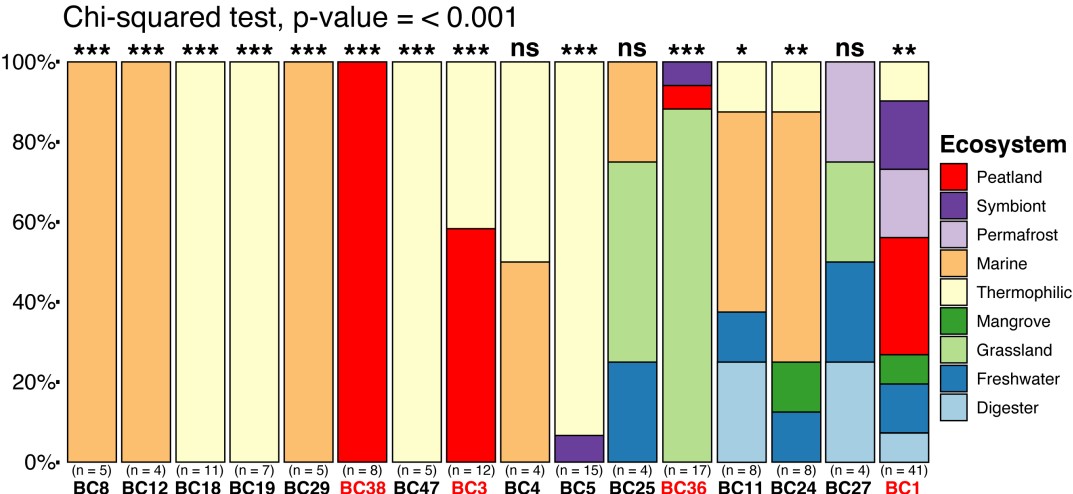

**FIG 2** Mosaic plot of the frequency of MAGs recovered by ecosystems in BCs with more than four representatives. BC labels in red represent clades with PMFB MAG representatives. Asterisks above each column represent the significance level of the relationship of an ecosystem to BC. Significance scale is as follows: *** - (0,0.001), ** - (0.002,0.01), * - (0.01,0.05), ns, not significant.

content, which, to our knowledge, differs from what is a common trait of thermophilic archaea (61, 62). Distribution of BCs by ecosystem and significant differences in GC content between BCs suggest divergent evolutionary trajectories across clades and potential for distinct functions in the Bathyarchaeia.

## Metabolic comparison and putative differences between PMFB clades (and associated MAGs)

To investigate the relationship between phylogenetic distance and functional potential within BC clades containing PMFB MAGs, gene clusters from a meta-pangenomic analysis were annotated using a combination of eggNOG (v5) (49) and the nr database (50). We observed a weak correlation between phylogenetic distance and functional dissimilarities between the nine BCs tested (Mantel statistic r: 0.03198 *P*-value < 0.05). We detail common metabolic pathways and distinct functions between BC1, BC3, and BC38-41, which include multiple PMFB MAGs (singleton BCs, BC15, and BC42 were excluded from this step), in the following sections and additionally recognize BC36 as a distinct cluster with one PMFB MAG but comprised primarily of other environments. We build upon and refine previous metabolic models of these groups (52) with specific emphasis on metabolism found within the PMFB.

### *Carbon metabolism*

Within the evaluated clades, most MAGs possess a partial Embden-Meyerhof-Parnas (EMP) glycolysis pathway but lack hexokinase (*glk*), which is used for the initial phosphorylation of glucose (File S3). Loh et al. (19) have suggested that due to the incomplete EMP pathway and the presence of phosphoenolpyruvate synthetase (*pps*) and fructose-1,6-bisphosphate aldolase (*fbaAB*) (FBP), these genes function exclusively in gluconeogenesis. Phylogenetic analysis shows that most MAGs within BC1, BC3, BC36, and BC38-41 have a class 2 FBP, whereas some from BC1, BC3, and one from BC40 have a class 1 FBP (Fig. S4A). Class 1 FBP activity can be induced in *Escherichia coli* when grown on gluconeogenic carbon substrates, whereas class 2 is constitutively expressed and indicative of a primary use with glycolytic substrates (63). The presence of a class 1 FBP in only some populations of BC1, BC3, and BC40 suggests that a gluconeogenic function is less common in *Bathyarchaeia*. Sugar-phosphate utilization may be possible in class 2 FBP-containing populations from BC1, BC3, BC36, and BC38-41, although sugar transporters could not be identified. Although different isoforms of FBP are found

within *Bathyarchaeia,* experimental evidence is required to identify if sugar utilization is possible.

BC36 lacks phosphofructokinase (*pfk*) but encodes the full glyoxylate pathway, an alternative route for sugar-phosphate utilization in Bathyarchaeia (15). Findings in Haloarchaea (64, 65) suggest that the glyoxylate pathway and incomplete EMP function in neither anaplerotic nor gluconeogenic pathways, but rather used for growth on acetate. Carbohydrate utilization varies greatly across BCs, with BC38-41 putatively using EMP for gluconeogenic purposes.

Alternative to carbohydrate utilization, many MAGs from BC38-41 possess the genetic potential for acetoin degradation (Fig. 3). This is evidenced by the presence of genes *acoL*, *acoC*, and *acoAB*, which facilitate the conversion of acetoin into acetaldehyde and acetyl-CoA. Furthermore, in 65% of BC38-41 MAGs, the oxidation of propanol to acetaldehyde might be possible via *adhP,* an alcohol dehydrogenase with the preference for propanol. The resulting acetaldehyde, formed either from acetoin or propanol, could undergo further conversion to acetate via an aldehyde dehydrogenase present in MAGs from BC38 and BC40. The likelihood of alcohol fermentation in BC38-41 is further supported by the presence of an acetyl-CoA synthetase (*acs*). We note that acetate formation from acetyl-CoA is a common feature in most MAGs from the seven BCs.

## Carbon fixation

Autotrophic lifestyles appear common among the Bathyarchaeia. The archaeal Wood–Ljungdahl (WL) pathway was found in most MAGs from BC1 and BC3 (File S3). However, previous reports have suggested that the absence of the *fmdE* subunit in the Bathyarchaeia Formyl-MFR dehydrogenase complex is characteristic of this group (19); nonetheless, our findings suggest this to be the case only for BC1 because in BC3, a homolog of *fmdE* was detected. The lack of Methylene-H$_4$MPT reductase (*mer*) in 13% of BC1 or BC3 MAGs supports proposals (19) that deeper branching Bathyarchaeia have lost

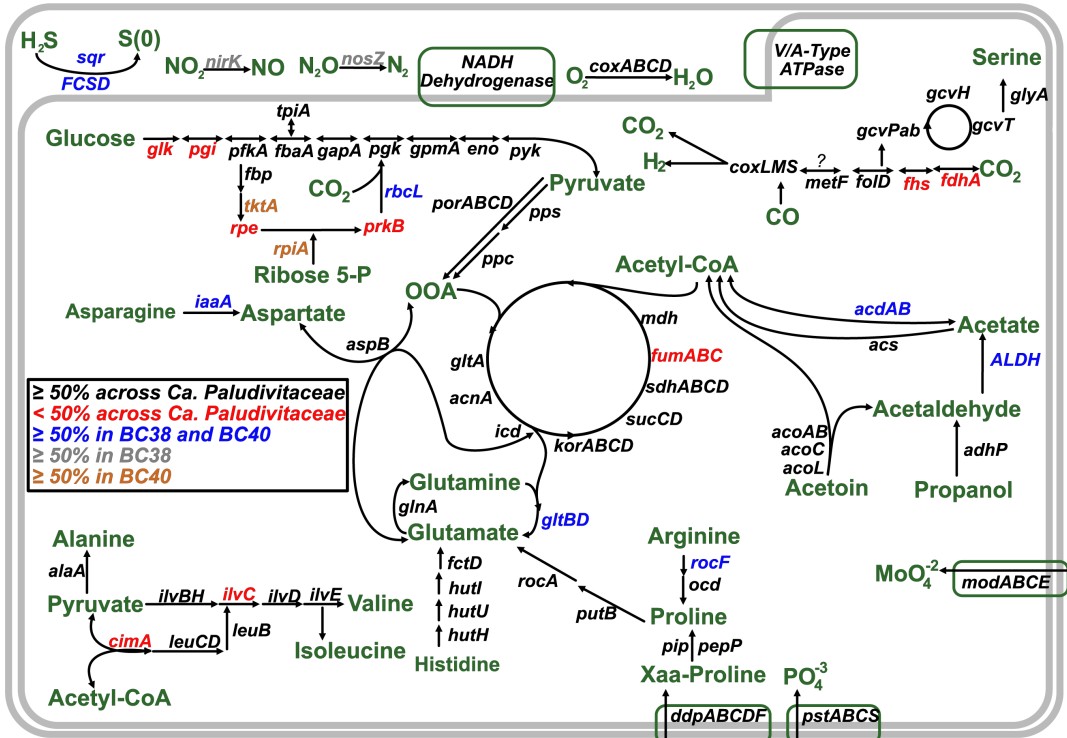

**FIG 3** Proposed key (high frequency) and variable metabolic characteristics in *Ca*. Paludivitaceae. Gene names are color-coded by the frequency of MAGs with that gene by BC. Key intermediates and pathway outputs are displayed in green. Question mark symbol represents an unknown function. A detailed list of corresponding KEGG numbers, gene names, and presence/absence in respective MAGs are listed in File S3.

the capacity to reduce $CO_2$ all the way to the methyl level. The carbonyl branch of the WL pathway is carried out by the CO dehydrogenase complex (*cdhABCE*) that condenses CO with a methyl group and CoA to form acetyl-CoA. The *cdhABCE* complex was found in almost all MAGs from BC1 and BC3.

In comparison to the CO dehydrogenase complex present in BC1 and BC3, all but one MAG from BC38-41 have putative aerobic molybdenum containing CO dehydrogenase, *coxMSL* capable of producing $CO_2$ and $H_2$ (Fig. 3). A phylogenetic analysis of the large subunit (*coxL*) indicates that BC38-40, clusters closely with other thermophilic CO-oxidizing archaea (66, 67) (Fig. S5). We note that we could not identify a *coxL* homolog in MAGs from BC41, but they did have the other two subunits *coxM* and *coxS*. High concentrations of CO, approximately 4 mM, have been shown to inhibit methanogenesis (68). These concentrations are not uncommon in natural environments (69) and suggest a potential for BC38-41 populations to alleviate CO stress and provide metabolites to peatland methanogens. As a byproduct of methanogenesis CO could be detoxified by BC38-41 and in turn produce compounds $CO_2$ and $H_2$ that could be used to fuel methanogen metabolism.

Alternatively, CO oxidation can be coupled to $CO_2$ fixation (70) through the Calvin cycle via ribulose-1,5-bisphosphate carboxylase (rbcL) which was present in clades BC38, BC39, and BC40. However, only two MAGs have the genetic potential for the near full cycle, lacking only phosphoribulokinase (*prkB*). *Archaeoglobus fulgidus* grows on CO with formate as an intermediate (71), via a novel type of formate dehydrogenase (FDH); however, only two of 13 *BC38-41* MAGs have FDH homologs. Instead, BC36 and BC38-41 have both methylene-$H_4$F reductase (*metF*) and methenyltetrahydrafrofolate cyclohydrogenasese (*folD*), involved in the bacterial WL pathway, which suggests the potential for formate utilization. Although no formate transporters were found, it is plausible that *BC38-41* can convert formate to 5,10-methylene-THF and subsequently feed into the glycine cleavage system.

### Oxygen, nitrogen, and sulfur metabolism

*BC38-41* are facultative anaerobes inhabiting water-logged peatland ecosystems. The low-affinity $O_2$ cytochrome aa3 oxidase was present in most of the MAGs in BC38-41, with four having all subunits (Fig. 3). This is consistent with our findings that *BC38-41* can putatively fix CO aerobically. Bathyarchaeia has only been suggested to tolerate aerobic conditions (72), but our findings suggest that BC38-41 may prefer to occupy oxygen-rich niches, which is in agreement with the recent analysis from (52).

Existence in anaerobic soil by BC38-41 is also possible through alternative chemotrophic mechanisms. MAGs from BC36 and BC38 encode a flavocytochrome sulfide dehydrogenase (FCSD), and BC40 harbors a type 3 sulfide:quinone oxidoreductase (SQR) (Fig. S4B). This suggests that these two clades in BC38-41 can potentially utilize $H_2S$ as an electron donor, facilitating sulfide-dependent respiration (73). Bathyarchaeia have been found in sulfur-rich environments (74, 75), and they may be playing a direct role in sulfur cycling in PMFB peatlands. N reduction, via *nosZ* or *nirK*, was identified in many MAGs from BC36 and BC38, but only in one MAG in BC39 and two in BC40. Direct fixation of N was only found in three MAGs from BC1 and BC41 and is likely a sparse function. Contrary to the report of (16), a significant contribution of $NH_4$ by Bathyarchaeia is not a common characteristic. The respiratory potential to occupy either aerobic or anaerobic soil conditions is common in BC38-41 MAGs. These adaptations are well suited for changes in $O_2$ availability brought on by the seasonality of flood-driven tropical peatlands (2).

### Amino acids and transport

Amino acids have been suggested as a primary carbon source for the Bathyarchaeia (17). However, this appears unlikely for BC1, which has mostly complete *de novo* biosynthesis pathways for 11 amino acids (Fig. 3). Additionally, BC1 has a low gene copy number of amino acid transporters relative to the other BCs (Fig. 4). Peptide/Ni$^+$ and amino

acid transporters were observed to be 2-fold more abundant in *BC38-41* compared with the other clades (Fig. 4; File S3). MAGs from BC3, BC36, and BC38-41 have the genetic potential for histidine to glutamate conversion; however, in most BC38-41, we find gene copies for glutamate dehydrogenase allowing for the conversion of glutamate to oxaloacetate. The gene copy number for *aspB* (involved in the interconversion between aspartate and oxaloacetate) was found 2-fold higher in BC38-41. In addition, BC38-41 also harbors *iaaA*, which catalyzes the conversion of asparagine to aspartate. BC38-41 encodes two cytosolic peptidases (*pip* and *pepP*) with the capacity to cleave proline from imported peptides. The liberated proline may be converted to glutamate, as evidenced by the presence of both *putB* and *rocA*. Collectively, this suggests that BC38-41 are adapted for growth on peptides, for either biosynthetic requirements and/or gluconeogenic purposes, with oxaloacetate serving as a key intermediate metabolite in their metabolism. The PMFB Bathyarchaeia BC38-41 harbor genomic evidence for the utilization of proline. Proline accumulation in plants is a common response to abiotic stressors such as extreme heat and drought (76). Projections of increased drought frequency and heat severity in the PMFB (11) suggest that local vegetation may accumulate more proline in response to these conditions. When these stressed plants die and decompose, soil proline levels will increase providing a selective advantage and favor the growth of BC38-41 populations.

The high-affinity phosphate transport system (*ptsABCS*) was universally present in all BCs, indicating an adaptation for low phosphate environments. BC1 has many transports for various inorganic ions, with a high prevalence of $Fe^{3+}$ transporters (Fig. 4). In summary, a diverse array of nutrient acquisition and metabolic capabilities are exhibited among BCs, with BC38-41 potentially relying on the presence of amino acids within their niche.

## Methanogenesis

We recovered many genes that are accessory to methanogenesis, such as *hrdABCD*, *acdAB*, *mchADG*, and *mtrH* in all clades excluding BC36 and BC39 (Supplementary File S3). However, no homologs for mcrA or "MCR-like" genes were detected in any of the MAGs from this study. These findings suggest that a functional methanogenic pathway in the Bathyarchaeia is unlikely in tropical peatlands (19, 26, 77).

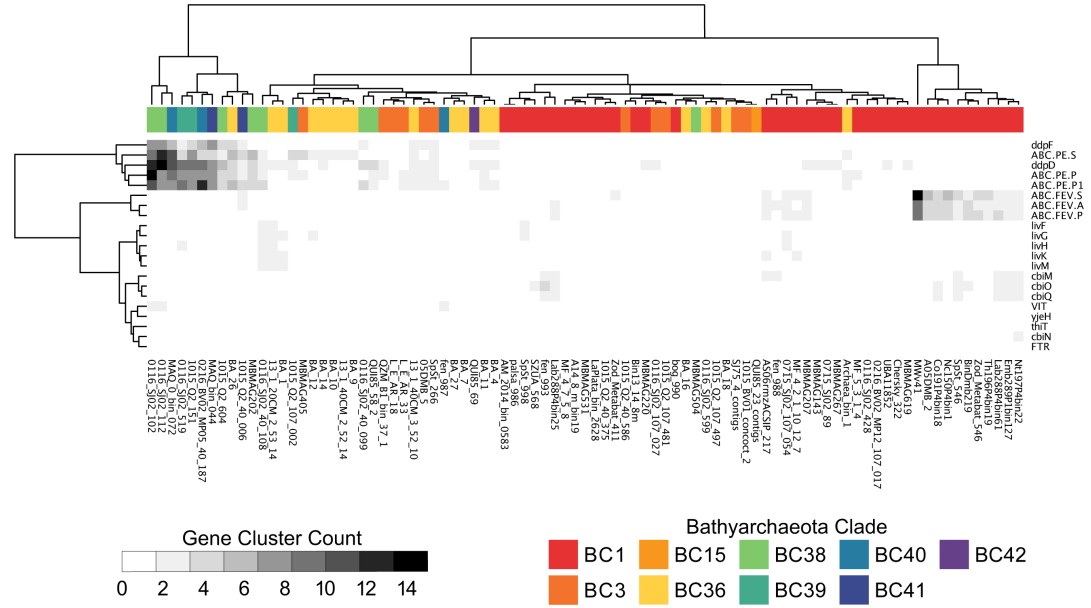

**FIG 4** Analysis of the abundance of cations and amino acid transport gene clusters. Clustering analysis was completed for gene cluster frequency (side dendrogram) and the phylogenetic affiliation of MAGs is depicted in the top dendrogram. The colored bar on the top indicates the BC of MAG.

## Metapangenomic analysis of PMFB Bathyarchaeia clades

To extend our understanding of the functional partitioning between tropical peatlands' Bathyarchaeia, we conducted a metapangenomic analysis on the nine BCs containing MAGs from the PMFB. Anvi'o identified 50,379 gene clusters across all BCs, with 19,338 found in at least one MAG from PMFB (Fig. 5). Due to the fragmented nature of MAGs and the evolutionary distance between BCs in analysis, no gene clusters were found in all PMFB MAGs. Only one gene cluster was found in 82 MAGs (88%) and another in 25 MAGS (89%) from the PMFB. Then, under this status, the metapangenome of all MAGs from the PMFB represents an open pangenome whose analysis can mainly provide general trends (Fig. S6).

The Bathyarchaeia metapangenome identified a distinct distribution of gene clusters within the relaxed-core of PMFB BCs (Fig. 5, with Table S3 detailing each cluster membership and annotation). Gene clusters were highly conserved within BC1 and BC3 MAGs, whereas BC38-BC41 showed a comparable high prevalence of conserved genes within their cluster evidencing a noticeable gene pool separation of these clades. BC15 and BC42, which are represented by a single MAG from the PMFB, share only a third of their gene clusters (not including singletons), 41% and 28%, respectively, with at least one other PMFB MAG. More than half of the gene clusters in the relaxed-core and shell of BC36 were exclusive to this clade pointing to its uniqueness, although deeper sampling can better test this observation. The relaxed-core gene clusters for all BCs accounted for 0.3%–21% of gene clusters found in each of the nine clades. The small percentage of gene clusters categorized as relaxed-core is analogous to previously reported thresholds at the class level (79). However, given our chosen completeness cutoff of >50%, we assume that many gene clusters categorized as shells are likely core genes misclassified because of the effects of assembly or binning errors. This assumption would bring the BC relaxed-core gene clusters to thresholds that are consistent with those reported for pan-genomes at the genus level (44, 80–82).

## Proposal of BC38-41 as *Candidatus* Paludivitaceae

Multiple lines of evidence in this study show that clades BC38-41 represent a distinct and cohesive group of Bathyarchaeia with unique genomic and metabolic features that seem

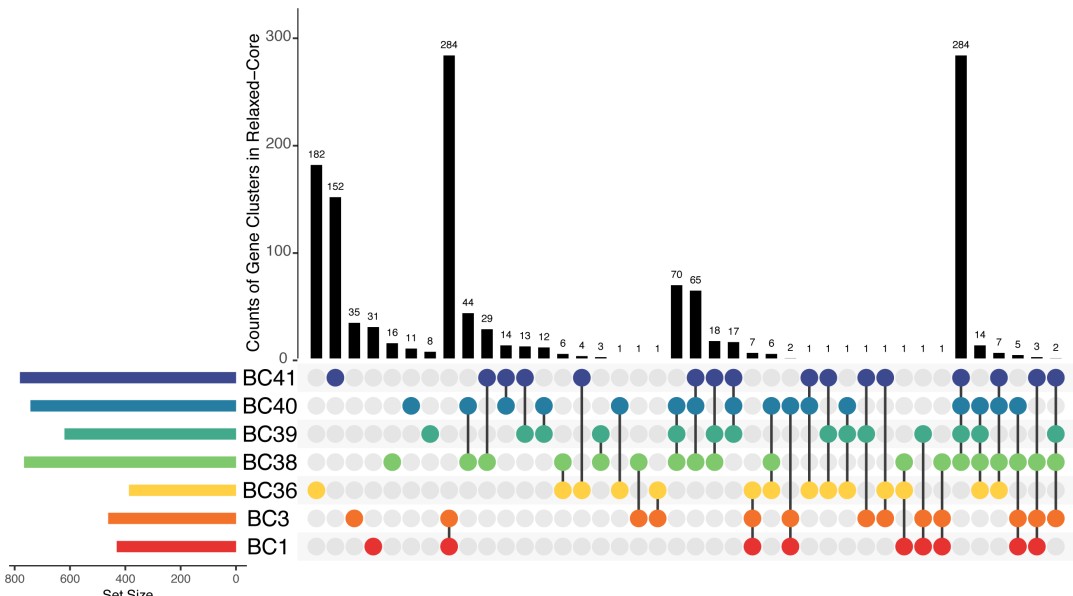

**FIG 5** Meta-pangenome analysis of selected BC clades: BC1, BC3, BC36, BC38, BC39, BC40, and BC41. Upset plots (78) show the distribution (upper black columns) and overlap of gene clusters between BCs (connected circles) found within the relaxed-core of clades (indicating the size of the set of genes in lower left bars).

specifically adapted to peatland environments. Several characteristics set them apart from other Bathyarchaeia and justify the proposal of a putative novel family within the phylum abundant in Amazon peatlands (Bathyarchaeia has an average read mapping frequency of 4.6%, QUI:6.9%, and SJO: 7.1% in BVA, QUI, and SJO metagenomes, respectively). Genomic data for BC38-41 show that AAI scores between all 16 MAGs average 64.3%, which is below the cutoff for genus-level classification (46). Furthermore, these clades display minimal variation in GC content (46.7%–51.8%) and coding density (84.3%–90.4%). The MAGs are phylogenetically grouped based on 54 single-copy genes within a clade supported by a bootstrap value of 100. In addition, there are 1,135 gene clusters shared only among the 21 MAGs highlighting their relatedness.

Metabolically, BC38-41 would use the EMP for gluconeogenic purposes with the potential to degrade acetoin or ferment propanol. Members are carboxydotrophic and putatively capable of carbon fixation through a proposed pathway of CO reduction to the Calvin cycle. In addition, BC38-41 are facultative anaerobes, having the respiratory potential to occupy both aerobic (*coxABCD*) and anaerobic (*nirK*, *nosZ*, and SQR) niches that may fluctuate seasonally in flooding tropical peatlands. Moreover, BC38-41 seem to be adapted for growth on peptides, particularly peptides with a terminal proline, given their high gene copy number of amino acid transporters and interconversions that lead to oxaloacetate.

This putative family of Bathyarchaeia has been recently proposed as in the order Houtuarculales (55), but given their persistent recovery from peatland environments and their potential unique metabolic adaptations for this type of environment, we delineate and propose the BC38-41 as the novel putative family *Candidatus* Paludivitaceae. The standing taxonomic name of this BC or *Candidatus* family (or order) is to be established along with the future isolation of a culture representative. The *Ca*. Paludivitaceae's name is derived from the Latin words "palus" referring to marsh or swamp environments and "vita" meaning life. Together, this implies the commonality of both ecological dwelling and metabolic traits suited for life found in peatland environments.

## Conclusions

Bathyarchaeia are abundant within PMFB soils; however, our knowledge of their metabolic potential and ecological functions is limited. Here, we detail the presence of nine clades within the Bathyarchaeia in PMFB. Genomic evidence points out that key groups (BC38-41 or *Ca*. Paludivitaceae) are facultative anaerobic carboxydotrophs capable of conserving energy through the aerobic oxidation of CO. Moreover, they are mixotrophic and able to generate energy from organic compounds such as acetoin, propanol, or peptides with terminal prolines. The *Ca*. Paludivitaceae has also the genomic potential to use CO for both energy and biomass in aerobic environments. These various metabolic findings propose that *Ca*. Paludivitaceae can play a significant role through various interactions including those with methanogenesis in the carbon cycle of tropical peatland environments.

## ACKNOWLEDGMENTS

This material is based upon work supported by the National Science Foundation under grant no. CAREER-1749252 to H.C.-Q. Work (proposal: 10.46936/10.25585/60000849) conducted by the U.S. Department of Energy Joint Genome Institute (https://ror.org/04xm1d337), a DOE Office of Science User Facility, is supported by the Office of Science of the U.S. Department of Energy operated under Contract No. DE-AC02-05CH11231.

M.J.P. – data curation, data analyses, data visualization, and validation, writing original and editing draft; A.I.G. – data analysis, review & editing draft; S.A. - data analysis, review & editing draft; F.M.-T. – field and molecular data collection, review & editing draft, R.T.-E. - field and molecular data collection, review & editing draft; H.C.-Q. – conceptualization and data collection

## AUTHOR AFFILIATIONS

[1]School of Life Sciences, Arizona State University, Tempe, Arizona, USA

[2]Swette Center for Environmental Biotechnology, Biodesign Institute, Arizona State University, Tempe, Arizona, USA

[3]Center for Fundamental and Applied Microbiomics, Biodesign Institute, Arizona State University, Tempe, Arizona, USA

[4]Laboratory of Soil Research, Research Institute of Amazonia's Natural Resources, National University of the Peruvian Amazon, Iquitos, Loreto, Peru

[5]School of Forestry, National University of the Peruvian Amazon, Iquitos, Loreto, Peru

## AUTHOR ORCIDs

Hinsby Cadillo-Quiroz http://orcid.org/0000-0002-4908-4597

## FUNDING

| Funder | Grant(s) | Author(s) |
| --- | --- | --- |
| National Science Foundation (NSF) | 1749252 | Hinsby Cadillo-Quiroz |
| Joint Genome Institute (JGI) | 10.46936/10.25585/60000849 | Hinsby Cadillo-Quiroz |

## AUTHOR CONTRIBUTIONS

Michael J. Pavia, Data curation, Formal analysis, Methodology, Validation, Visualization, Writing – original draft, Writing – review and editing | Arkadiy I. Garber, Formal analysis, Writing – review and editing | Sarah Avalle, Formal analysis, Writing – review and editing | Franco Macedo-Tafur, Data curation, Resources, Writing – review and editing | Rodil Tello-Espinoza, Data curation, Resources, Writing – review and editing | Hinsby Cadillo-Quiroz, Conceptualization, Data curation, Funding acquisition, Methodology, Project administration, Resources, Supervision, Writing – original draft, Writing – review and editing

## DATA AVAILABILITY

MAG analysis code is available at https://github.com/Hinsby/BathyarchaeaMAGs2023.

## ADDITIONAL FILES

The following material is available online.

### Supplemental Material

**Data S1 (Spectrum00387-24-S0001.xlsx).** Metagenomes metadata, assembly, and MAGs distribution.
**Data S2 (Spectrum00387-24-S0002.csv).** Details, classification, and accession information of MAGs in study.
**Data S3 (Spectrum00387-24-S0003.xlsx).** Annotated gene analyses across MAGs.
**Supplemental material (Spectrum00387-24-S0004.pdf).** Tables S1 to S3; Fig. S1 to S6.

### Open Peer Review

**PEER REVIEW HISTORY (review-history.pdf).** An accounting of the reviewer comments and feedback.

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
