## [Reviewer comments · Microbiology Spectrum]

Microbiology Spectrum

Functional insights of novel Bathyarchaeia reveal metabolic versatility in their role in Peatlands of the Peruvian Amazon

Michael Pavia, Arkadiy Garber, Sarah Avalle, Franco Macedo-Tafur, Rodil Tello-Espinoza, and Hinsby Cadillo-Quiroz

Corresponding Author(s): Hinsby Cadillo-Quiroz, Arizona State University

Review Timeline:

Submission Date:	February 10, 2024
Editorial Decision:	July 2, 2024
Revision Received:	October 18, 2024
Accepted:	October 24, 2024

Editor: Katharina Kujala

Reviewer(s): Disclosure of reviewer identity is with reference to reviewer comments included in decision letter(s). The following individuals involved in review of your submission have agreed to reveal their identity: Bing Song (Reviewer #1)

Transaction Report:

DOI: <https://doi.org/10.1128/spectrum.00387-24>

Re: Spectrum00387-24 (Functional insights of novel Bathyarchaeia reveal metabolic versatility in their role in Peatlands of the Peruvian Amazon)

Dear Dr. Hinsby Cadillo-Quiroz:

Thank you for the privilege of reviewing your work. Below you will find my comments, instructions from the Spectrum editorial office, and the reviewer comments.

As you will see, both reviewers were positive about your manuscript but have suggested minor modification to further improve its quality.

Revision Guidelines

Sincerely,
Katharina Kujala
Editor
Microbiology Spectrum

Reviewer #1 (Comments for the Author):

Although the manuscript is well-written, authors still need to double-check for small errors such as spelling, citations, and so on. Additionally, genome assembly and binning might be too simplistic. Authors could consider using another assembly tool, such as MetaSPAdes, and more binning tools like binny and VAMB, to obtain more reliable high-quality MAGs.

Reviewer #2 (Comments for the Author):

Summary of Key Findings

This study seeks to elucidate the metabolic diversity of understudied Archaea, specifically Bathyarchaeia, in peatlands with varying nutrient regimes and vegetation types. The lack of cultured representatives makes a metagenomic approach necessary to start understanding these abundant members in peatland and other terrestrial and aquatic communities. The authors utilized publicly available MAGs as well as new assemblies to more broadly characterize clades of Bathyarchaeia (BCs), specifically core metabolic processes from carbon to methanogenesis. With this data they propose a new Candidatus family, Candidatus Paludivitaceae, for a peatland-adapted clade. The analyses are well done and clearly communicated. I have some minor concerns for the authors to address below.

Minor Concerns:

The authors have described their -omics analysis clearly with good citation. However, there is no link to code availability (i.e. through GitHub or similar). It may also be helpful to include the BioProject number in the main text for the BioSamples included in the Supp. Table.

Line 112: Please clarify why only MAQ metagenomes were co-assembled. Or, why the other sites weren't.

Line 115: Why were the >50% and <10% completed/redundant criteria utilized? Did this represent a clear threshold delineation in your dataset?

Lines 121-122: Edit for clarity.

Line 149: What was the version of the nr database (or what accession date)?

Lines 338-339: Are the authors suggesting that proline is a source from plants in this environment, or rather, that there may be a similar mechanism for proline accumulation in archaea? I'm not sure of the relevance of the plant proline reference here.

Fig. 1: Label in plot for genome size and GC content is not aligned with data layers around the phylogeny. I would suggest aligning these.

Fig 2: Legend should indicate significance of red vs black text for BC IDs (i.e. why are BC28, 3, 36, and 1 in red text?)

Fig. 5: Are the gene clusters here related to the supplemental info on Pangenome analysis data? Is there a way to annotate the gene functions in this plot for these clusters, for example, adding labels to the x-axis below the connected circles? Or alternatively, color code the currently black bars to relate to the supp info on these relaxed-core gene clusters? It seems this would help in your ability to interpret and discuss some of your pangenome findings.

Fig. S1: plot axes are present but data points are missing.

Fig. S2-S5: text is so small it limits interpretation. Make sure files are uploaded with larger size for easier interpretation.

Thank you for providing me with the opportunity to review the research article titled "Functional insights of novel Bathyarchaeia reveal metabolic versatility in their role in Peatlands of the Peruvian Amazon". In this study, the authors investigate the metabolic versatility of 28 metagenome-assembled genomes (MAGs) belonging to a putative novel family, "Candidatus Paludivitaceae", endemic to peatlands. Their findings suggest that the MAGs of Ca Paludivitaceae possess the genetic potential for oxygen, sulfide, or nitrogen oxidation, while clades outside Ca Paludivitaceae are putatively capable of acetogenesis, de novo amino acid biosynthesis, and encode a high number of Fe³⁺ transporters. The presentation of these descriptions portrays an environmentally friendly microorganism for readers. However, it is imperative that the authors address the following comments to enhance the manuscript's quality. Section-wise comments are listed below:

Abstract:

1. From the abstract, perhaps one more sentence is needed to introduce the necessity of focusing on the Ca Paludivitaceae MAGs?
2. In the abstract, it appears that you assembled novel Bathyarchaeia MAGs and described their functions. Therefore, the novelty of the paper lies in assembling novel genus-level MAGs and elucidating their unique functions, which have not been discovered by others?

Introduction:

1. Line 51, 52, 53, 54: What does (2), (2) (3), (3) (4), (4) (5), and (5) mean? Maybe something wrong with the citation format? Better check all citation format in the manuscript.

Material and methods:

1. In the binning process, only one binning tool -- MetaBAT2 was used. Why not try more, such as new tools binny and vamb to get more high quality and reliable HQ MAGs? And, for the completeness of MAGs, author can try to use to check rRNA using .
2. Line96: spelling error --- assesara aaaaaambly ???
3. Line 127: completeness higher than ? and contamination lower than ?

Results

1. Line155: Microbial Community Composition from assemblies of new MAGs --- Pay attention to the capitalization of each word in the title.
2. Line163-165: The consistency of overall recovery of MAGs with previous studies have been described, so what are the differences with before? The consistency of the overall recovered MAGs with previous studies has been described. However, it may also be important here to emphasize the differences, aside from the similarities.

Figure

1. Authors should format the text within figures according to the submission instructions.

We thank the reviewers their insightful comments and points. Below, find a point-by-point answers document.

Reviewer: 1

- Comments to the Author

Although the manuscript is well-written, authors still need to double-check for small errors such as spelling, citations, and so on. Additionally, genome assembly and binning might be too simplistic. Authors could consider using another assembly tool, such as MetaSPAdes, and more binning tools like binny and VAMB, to obtain more reliable high-quality MAGs.

 - We thank the reviewer for positive feedback and helpful suggestions. We indeed have corrected minor errors (spelling, punctuation, citation) and carefully reviewed our methods to ensure the reliability and quality of our assembly and binning. Evolving approaches can provide MAG improvements, but we have also paid attention to staying close to conservative interpretations of findings, acknowledging that MAGs can always be reanalyzed.
- From the abstract, perhaps one more sentence is needed to introduce the necessity of focusing on the Ca Paludivitagea MAGs?
 - Thank you for your comment. We have added a sentence to illustrate better why we focus on Ca. Paludivitageae. Now at L35-36 it reads “*We focus on the Ca. Paludivitageae MAGs due to the novelty of this group and the limited understanding of their role within tropical peatlands.*”
- In the abstract, it appears that you assembled novel Bathyarchaeia MAGs and described their functions. Therefore, the novelty of the paper lies in assembling novel genus-level MAGs and elucidating their unique functions, which have not been discovered by others?
 - Thanks for pointing out the need to highlight the importance and frame of our work. We have updated the abstract to include an importance section, required by journal, where we have highlighted the novelty of our work. This section begins on L49-L56
- Introduction:
- Line 51, 52, 53, 54: What does (2), (2) (3), (3) (4), (4) (5), and (5) mean? Maybe something wrong with the citation format? Better check all citation format in the manuscript.
 - Thank you for your observation regarding the repeating number format on that section. The first numbers in parentheses were used to guide readers through the list of factors contributing to the slow process of organic matter decomposition, and it coincidentally matched the associated citation numbers. We have removed these guiding numbers to avoid any confusion. All citation formats have also been checked and corrected accordingly in L61-66.
- Material and Methods:
- In the binning process, only one binning tool -- MetaBAT2 was used. Why not try more, such as new tools binny and vamb to get more high quality and reliable HQ MAGs? And, for the completeness of MAGs, author can try to use to check rRNA using .
 - Thank you for your valuable feedback. In our previous publication (Pavia et al., 2023), we described an extensive binning pipeline where we employed 4 binning tools and found that MetaBAT2 was the most effective at recovering MAGs from our environment and type of sequencing platform (Illumina). In addition to MetaBAT2, we also used MaxBin2 but the resulting MAGs were of significantly

lower quality, which led us to decide against consolidation with DASTool. We did use Barnap to check for rRNA genes; however, none were detected in our MAGs. We acknowledge that using additional new binning tools like binny and VAMB could improve the quality of MAGs, but we did not find dramatic changes in a test set, perhaps because the MAGs we focused on are a fraction of the data (25%); nevertheless, we will consider incorporating these tools in our future studies.

- Line96: spelling error --- assesara aaaaaambly ???
 - Apologies for this strange mistyping. Text has been corrected in now L108.
- Line 127: completeness higher than ? and contamination lower than
 - Thank you for pointing out the confusing wording of the sentence. We have edited the sentence to explain better how MAGs were chosen. Now in L141 – 143 reads “*In instances where more than one MAG from the same environment had an AAI greater than 95%, we considered them replicates and selected the MAG with higher completeness and lower contamination.*”
- Results:
- Line 155: Microbial Community Composition from assemblies of new MAGs --- Pay attention to the capitalization of each word in the title
 - Thank you for pointing this out. We have corrected the capitalization scheme in this subtitle in now L170.
- Line 163-165: The consistency of the overall recovered MAGs with previous studies has been described. However, it may also be important here to emphasize the differences aside from the similarities.
 - Thank you for your comment. We have changed the sentence to help with clarity. The studies we referenced used 16S amplicon sequencing only, so a direct comparison cannot be made. We understand how this was confusing in our writing and have adjusted L178 – 180, which now reads “*This study extends these findings by recovering HQ and MQ MAGs, providing a more detailed genomic representation of the Bathyarchaeia populations within peatlands from QUI, SJO, and MAQ, from the PMFB.*”
- Authors should format the text within figures according to the submission instructions.
 - Thank you for pointing this out. We formatted the text within all the figures according to the submission instructions.

Reviewer: 2

- The authors have described their -omics analysis clearly with good citation. However, there is no link to code availability (i.e. through GitHub or similar). It may also be helpful to include the BioProject number in the main text for the BioSamples included in the Supp. Table.
 - Thank you for pointing this out. To make code available, we have made available the code steps as suggested in github repository, however since the BioSamples are already in supplementary materials we believe information can be accessible from there without trouble and keeping MS compact.
 - L133-134 now includes: " *MAG analysis code is available at <https://github.com/Hinsby/BathyarchaeaMAGs2023>.*"
- Line 112: Please clarify why only MAQ metagenomes were co-assembled. Or, why the other sites weren't.

- We appreciate your comment and the opportunity to clarify. MAQ metagenomes were co-assembled because triplicate metagenomes (each with technical duplicate DNA and sequencing) were available for analysis. In contrast, only one metagenome each from SJO and QUI was available from the triplicate samples due to the difficulty of retrieving high-quality DNA soil from depths >60cm and the library yielding a low number of reads (low levels of DNA with inhibitory contaminants).
- This is now addressed in L124-126 with "*MAQ metagenomes were co-assembled by transect, while SJO and QUI were a single assembly due to low DNA and low sequencing yield of replicates.*"
- Line 115: Why were the >50% and <10% completed/redundant criteria utilized? Did this represent a clear threshold delineation in your dataset?
 - Thank you for the great question. We wish to stay in comparable values to other studies and thus follow the criteria of >50% completeness and <10% redundancy as a well-documented and widely accepted cutoff for ensuring the reliability of MAGs as described in Bowers et al., 2017 (reference #51)
- Lines 121-122: Edit for clarity.
 - Thank you for pointing this out. We have changed the sentence to improve its clarity. L136 - 138 now reads "*To determine the distribution of PMFB Bathyarchaeia within the class, we performed phylogenetic inference using an in-group of 238 Bathyarchaeia MAGs, which included both those from this study and publicly available ones*"
- Line 149: What was the version of the nr database (or what accession date)?
 - The nr database that we used was downloaded in August 2021. Now in L164 – 165 it reads "*were annotated using eggNOG (v5) (51) and the nr databased (accessed August 2021) (52) using a consensus sequence built from the alignment of each gene cluster.*"
- Lines 338-339: Are the authors suggesting that proline is a source from plants in this environment, or rather, that there may be a similar mechanism for proline accumulation in archaea? I'm not sure of the relevance of the plant proline reference here.
 - Thank you for directing our attention to this possibly confusing section. We intended to convey that proline concentrations will increase in the soil as a result of abiotic stressors on the plants. Proline levels are a known and used evaluation to indicate the level of stress resistance by plants. The increase in proline release would provide a selective nutritional advantage to BC38-41 populations. We have clarified these points in L351 – 356 which now reads "*The PMFB Bathyarchaeia BC38-41 harbor genomic evidence for the utilization of proline. Proline accumulation in plants is a common response to abiotic stressors such as extreme heat and drought (78). Projections of increased drought frequency and heat severity in the PMFB (11) suggest that local vegetation may accumulate more proline in response to these conditions. When these stressed plants die and decompose, soil proline levels will increase providing a selective advantage and favor the growth of BC38-41 populations.*"
- Fig. 1: Label in plot for genome size and GC content is not aligned with data layers around the phylogeny. I would suggest aligning these.
 - We agree and have adjusted these legends so that they align with the data layers.

- Fig 2: Legend should indicate significance of red vs black text for BC IDs (i.e. why are BC28, 3, 36, and 1 in red text?
 - Thanks for bringing this to our attention. We have changed the figure legend to explain why some BCs are red. Now, on L696, it reads, “*BC labels in red represent clades with PMFB MAG representatives.*”
- Fig. 5: Are the gene clusters here related to the supplemental info on Pangenome analysis data? Is there a way to annotate the gene functions in this plot for these clusters, for example, adding labels to the x-axis below the connected circles? Or alternatively, color code the currently black bars to relate to the supp info on these relaxed-core gene clusters? It seems this would help in your ability to interpret and discuss some of your pangenome findings.
 - Thank you very much for the suggestion. Here the pangenome analysis was used as a broad way to represent the relationship among Bathyarchaeia subgroups based on gene clusters; however, the first conclusion was that the data is rather fragmented, with no single cluster having a copy in all assessed MAGs (L373-374). Based on focused on the relaxed core relationships among clusters (Fig 5) and Table S3 provides the specific distribution of genes among clusters, their annotations, and also their frequency within SC. We have discussed and tested building up some of this info in Fig 5, but the output is an overcomplicated figure, and given the currently incomplete nature of the pangenome assessment, we propose to keep the structure of Fig 5 as is and point out that Table S3 details each cluster membership and annotation. L379-380 now includes “(*Figure 5, with Table S3 detailing each cluster membership and annotation*)”
- Fig. S1: plot axes are present but data points are missing.
 - Thank you, data points are confirmed there, when in pdf file format.
- Fig. S2-S5: text is so small it limits interpretation. Make sure files are uploaded with larger size for easier interpretation.
 - Thank you we have adjusted the text size of the supplementary figures.

Re: Spectrum00387-24R1 (Functional insights of novel Bathyarchaeia reveal metabolic versatility in their role in Peatlands of the Peruvian Amazon)

Dear Dr. Hinsby Cadillo-Quiroz:

Thank you for addressing all the comments made by the reviewer in the first round of revision. I am pleased to inform you that your manuscript has been accepted, and I am forwarding it to the ASM production staff for publication. Your paper will first be checked to make sure all elements meet the technical requirements. ASM staff will contact you if anything needs to be revised before copyediting and production can begin. Otherwise, you will be notified when your proofs are ready to be viewed.

Sincerely,
Katharina Kujala
Editor
Microbiology Spectrum